# Empirical Evaluation on Utilizing CNN-Features for Seismic Patch Classification

**Chunxia Zhang [1], Xiaoli Wei [1] and Sang-Woon Kim [2],[*]**

1   School of Mathematics and Statistics, Xi'an Jiaotong University, Xi'an 710049, China;
    cxzhang@mail.xjtu.edu.cn (C.Z.); wxl18847162006@stu.xjtu.edu.cn (X.W.)
2   Department of Computer Engineering, Myongji University, Yongin 17058, Korea
*   Correspondence: kimsw@mju.ac.kr

**Featured Application: Automatic seismic fault detection, exploration of underground resources, and other areas of image recognition where large-scale artificial data is possible but real-world data is extremely limited.**

**Abstract:** This paper empirically evaluates two kinds of features, which are extracted, respectively, with traditional statistical methods and convolutional neural networks (CNNs), in order to improve the performance of seismic patch image classification. In the latter case, feature vectors, named "CNN-features", were extracted from one trained CNN model, and were then used to learn existing classifiers, such as support vector machines. In this case, to learn the CNN model, a technique of transfer learning using synthetic seismic patch data in the source domain, and real-world patch data in the target domain, was applied. The experimental results show that CNN-features lead to some improvements in the classification performance. By analyzing the data complexity measures, the CNN-features are found to have the strongest discriminant capabilities. Furthermore, the transfer learning technique alleviates the problems of long processing times and the lack of learning data.

**Keywords:** seismic patch classification; CNN-features; transfer learning; data complexity

## 1. Introduction

Seismic faults are important subsurface structures that have significant geologic implications for hydrocarbon accumulation and migration in a petroleum reservoir. On the basis of these characteristics, it is very important to detect faults with advanced techniques. Recently, seismic fault detection using deep-learning techniques has been actively studied [1–5]. In this approach, seismic images are first divided into patches of a certain size. The fault detection problem then becomes a two-class classification problem that classifies fault and nonfault (normal) patches. The fault detection problem can be solved by identifying the location of the patches classified as abnormal patches in the fault line. This paper focuses on the classification of patch data. First, feature vectors were extracted from seismic patch data and were then classified as fault and nonfault patches using existing classifiers to find fault lines.

Convolutional neural networks (CNNs) are now state-of-the-art approaches for a lot of applications. On the basis of their good performance, CNNs have recently been used to detect seismic faults [4]. However, two constraints can be found in this approach: one is the need to provide a huge amount of interpreted data (e.g., fault and nonfault patches); the other is the significant amount of time required to process them. To address the first, a synthetic dataset, having simple fault geometries, was built. Therein, the input to the CNN was the seismic amplitude only; that is, the approach did not require the calculation of the other seismic attributes, but the second constraint remains without any solution.

As is commonly known, CNNs take a tremendous amount of time to learn when allowing "sufficient" training data. Recently, it has been observed that the convolutional

(C) and fully connected (FC) layers take the most time to run [6]. In particular, since the latter is responsible for the multiplication of large-scale matrices, it consumes up to almost 60% of the computation time. From the above review results, as well as from the findings in [6–11], we may consider replacing the FC layers responsible for classifying seismic patch data in a CNN with an existing classifier, such as support vector machines (SVMs). The role of the C layer in this framework corresponds to, for example, the role of the principal component analysis (PCA) in traditional statistical-based classification.

The use of SVMs instead of FC layers is known to improve the classification accuracy [7,12], but no analysis has been made on why. Rather than embarking on a general analysis, in this paper, we will consider a comparative study that will be taken as the basis for the above improvements. In addition, the measures of the data complexity can be used to estimate the difficulty in separating the sample points into their expected classes. Especially, it has been reported that the measurements can be performed to figure out a variety of characteristics related to data classification [13]. From this point of view, to derive an intuitive comparison and to answer the above question, we will consider the complexity measurements.

In this paper, we use CNN-features in seismic patch classification. Using this feature vector, we can avoid the problems of a long learning time and a lack of training data, without deteriorating the classification performance. We also analyzed the data complexity to compare the discriminating powers of the features. A preliminary short version of this paper was published as a conference paper [14]. In particular, the current version has been expanded to include: First, additional techniques to improve the classification performance of CNNs; second, a quadratic pooling strategy that was reviewed and tested in order to further increase the discriminative capabilities of the CNN-features [15]; and third, new experiments were conducted and analyzed to consider how improving CNNs in classification systems that classify CNN-features using conventional classifiers changes the system performance.

In this paper, we focus on the following research issues regarding the use of CNN-feature vectors. First, we attempt to hybridize deep-learning techniques and pattern recognition (PR) methods for seismic patch classification. Traditional PR proceeds in two stages: data representation and feature classification, while CNN is implemented in two stages: feature extraction and classification. It is well known that the representation phase of PR is sensitive to the domain properties, while the classification phase of a CNN has a long training time. The main issue of this paper is the hybridization of the above two stages to address these shortcomings. On the basis of this hybrid strategy, we can combine the feature extraction capabilities of CNNs with the various classifiers that have been developed so far and can utilize the appropriate classifiers for applications.

Second, we attempt to adopt transfer learning [16] in model learning for seismic patch classification. In general, while artificially generating experimental data is not expensive, there is a limit to presenting the detailed state of the real world. On the other hand, the real-world data can reflect the true state well, but it is difficult to obtain a sufficient amount. The second research point of the paper is the use of the transfer learning technique in the learning of models for the extraction of CNN-features. That is, we first learned CNN models using well-prepared artificial data, and then leveraged them as pre-trained models to learn classification models on real-world data.

The third issue of the paper is the application of data complexity in an effort to find reasons why this hybrid approach leads to improved classification results. To achieve our purpose, we measured the data complexity of the feature vectors extracted with traditional methods, including PCA, and compared them with that of the CNN-features.

Finally, the fourth concern of the paper is to represent the seismic patches in CNN-features. To avoid the curse of dimensionality in image classification, various approaches have been used so far. The CNN-features can be utilized regardless of their sensitivity to the cardinality and dimensions of the dataset since they are extracted from CNNs.

The remainder of the paper is organized as follows: in Section 2, a brief introduction to the latest findings on seismic fault detection, using CNN-features and data complexities, is provided; in Section 3, a classification method related to this empirical study and the structure of seismic patch image data are presented, in turn; in Section 4, experimental studies, such as experimental data and methods, and classification accuracy rates and data complexity measurements, are described in detail; in Section 5, conclusive arguments and limitations that are worthy of further research are summarized.

## 2. Related Work

This section briefly reviews some of the latest results for detecting seismic faults, using CNN-features and data complexities for measuring the discrimination power of the features.

### 2.1. Seismic Fault Detection

In geophysics, current fault detection methods can be broadly categorized into two classes: traditional methods [17–19], and machine-learning-based algorithms [1–5,20–23]. In general, traditional methods work by detecting the local discontinuity in seismic images on the basis of some of the seismic features, such as semblance, variance, etc. For example, Wang and Alregib [19] propose a combination of the Hough transform and the tracking vector to extract faults from coherence maps. Although these methods can produce satisfactory results in early studies, they are time-consuming, and they also require professional background knowledge to calculate the informative attributes.

With the great success of various machine-learning and deep-learning techniques in many fields, scholars have developed many automatic fault detection procedures to alleviate the shortcomings of traditional ones. For instance, SVMs [20] and the multilayer perceptron technique [21] are separately applied to analyze the attributes (including the detected edges, and the geometric and texture features) extracted from different seismic images, and, finally, to classify whether there are faults or not. In [5], Wang et al. provide a review of the use of image-processing and machine-learning methods to identify faults. Considering that the performance of machine-learning algorithms strongly relies on the calculated attributes, it is popular, in recent years, to employ deep-learning approaches to achieve automatic fault detection. Among them, the CNN is the most commonly used technique because of its powerful ability to extract useful features. Recently, it has been combined with transfer learning [1–3] in order to further enhance its performance while alleviating the difficulty in obtaining sufficient labeled data.

### 2.2. CNN-Features

CNN-features are composed of values taken from the activation unit of the first FC layer of the CNN architecture [8]. Various studies have been conducted using CNN-features in many applications. In [24], it was evaluated whether the CNN weights obtained from large-scale source tasks could be transferred to a target task with a limited amount of training material. Along with this study, many other studies related to the extraction and utilization of CNN-features have been conducted [6,9,10].

In [25], it was reported that reusing a previously trained CNN as a generic feature extractor leads to a state-of-the-art result, meaning that CNNs are able to learn generic feature extractors that can be used for different tasks. Thus, some studies have recently been reported in the industry on the techniques of extracting, and classifying feature vectors with this approach [7,11]. In [7], top-performing hand-crafted descriptors, including the LBP (local binary pattern) and the HOG (histogram of oriented gradients), were compared with CNN-based models using variants of AlexNet [26]. Experiments on three datasets reported that CNN-based models were superior to other models. However, it was also pointed out that, to extract meaningful features from raw data, the approach requires huge amounts of training data. In case the generation of data is expensive, it might not be appropriate, as in [27].

In the meantime, CNNs have improved performances while building deeper and wider networks [28]. In a different way, there are studies to improve the performance by simply using a pooling based on quadratic statistics, such as covariance, rather than sum (mean) or max pooling. A few examples are: second-order pooling (O2P) [15]; bilinear pooling (BiP) [12]; compact BiP [29]; deep architecture for O2P [30]; improved BiP [31]; matrix power normalized covariance (MPN-COV) [32]; and the iterative matrix square root normalization of covariance pooling (iSQRT-COV) [33].

In [12], an effective architecture, called a bilinear CNN (B-CNN), was developed for visual classification. B-CNNs represent an image as a pooled outer product of features, derived from two CNNs, which capture the localized feature interactions. The work in [31] shows that feature normalization and domain-specific fine-tuning provide additional benefits, improving the accuracy using identical networks. In subsequent studies, various methods were proposed, in turn, to overcome the disadvantages of the B-CNN, such as the high dimensionality of the feature vectors, or GPU-unfriendly algorithms.

### 2.3. Data Complexity

An attempt to identify the relationship between the data and the classifier that classifies it was specifically initiated in [34]. Since then, it has been studied a lot. A few examples, but not all, can be found in [13,35,36]. Referring to these findings, we selected the data complexity measures to be used in our experiments as the following four indicators: the Fisher's discriminant ratio ($F1$); the directional-vector Fisher's discriminant ratio ($F1v$, or simply, $Fv$); the distance of erroneous instances to a linear classifier, or its training error rate ($L2$); and the volume of the local neighborhood ($D2$).

Each measurement of complexity mentioned above is made as follows. First, $F1$ is used to measure the separation capacity of a single feature between two classes. One way to obtain the value for multiple features is:

$$F1 = \sum_{i=1}^{2} p_i(\boldsymbol{m} - \boldsymbol{m}_i)\boldsymbol{C}^{-1}(\boldsymbol{m} - \boldsymbol{m}_i), \tag{1}$$

where $\boldsymbol{m}_i$ (and $\boldsymbol{m}$) is the mean of each class (and $c_i$ the entire class); $\boldsymbol{C}$ is the pooled covariance matrix derived from averaging the covariance of each class; and $p_i$ is the proportion of examples in the class, $c_i$. As a result, the higher the value, the less redundancy and the easier it is to distinguish between the two classes.

Second, the $Fv$ value, developed as a complement to $F1$, is calculated as in [29]:

$$Fv = \frac{1}{1 + \boldsymbol{d}^T\boldsymbol{B}\boldsymbol{d}/\boldsymbol{d}^T\boldsymbol{W}\boldsymbol{d}}, \tag{2}$$

where $\boldsymbol{B}$ is the between-class scatter matrix; $\boldsymbol{W}$ is the within-class scatter matrix; and $\boldsymbol{d} = \boldsymbol{W}^{-1}(\boldsymbol{m_1} - \boldsymbol{m_2})$. That is, $\boldsymbol{d}$ is a directional vector in which the data are projected to maximize class separation, and the value of $Fv$ approaches zero by maximizing $\boldsymbol{W}^{-1}\boldsymbol{B}$, which becomes a simpler classification. Therefore, the lower the measurement, the simpler the classification problem is.

Third, $D2$ is related to the density measurement, defined as the average number of samples per unit volume in the space where all samples are distributed [36]. The value of $D2$ in $n$ training samples is determined by measuring the average volume occupied by the $k$ nearest neighbors in each sample, $x_i$, referred to as $N_k(x_i)$. The value is counted as:

$$D2 = \frac{1}{n}\sum_{i=1}^{n}\prod_{h=1}^{d}(max(f_h, N_k(x_i)) - min(f_h, N_k(x_i))), \tag{3}$$

where $max(f_h, N_k(x_i))$ and $min(f_h, N_k(x_i))$ denote the maximum and minimum values of the feature, $f_h$, among the k-NN of $x_i$, respectively.

Finally, the *L*2 value is decided by referring to the error rate of the linear SVM classifiers. Therefore, the higher the value, the more errors, and the more difficult it is to classify linearly, which increases the complexity. Here, each measurement is briefly described to the minimum required. A detailed description of each of these measurements, and other measures that were not selected here, but that were closely related to the selected ones, can be found in the relevant literature.

## 3. Methods and Data

This section first introduces the classification methods associated with the current empirical research, and then presents the seismic wave image data used in this paper.

### 3.1. Classification Methods

The classification system to be developed in this paper (named HYB) is a method of **hyb**ridizing an existing linear (or nonlinear) classifier (e.g., an SVM) with a CNN. HYB is a classification framework that performs feature extraction on CNNs, and then trains SVMs using the extracted features. In this approach, when learning the CNN in the target domain, the problem of insufficient learning data can be avoided through transfer learning, using a model learned in advance in the source domain.

The CNN architecture extracting the feature vector consists of an input layer, a hidden layer, and an FC layer. The hidden layer includes layers that perform convolution and subsampling. In this structure, the feature vectors we are trying to extract are made from input neurons directly connected to the FC layer. The extracted vector is a midlevel representation, just before the input image passes through the hidden layers and is transferred to the FC layer. Therefore, the dimension of the feature vector is equal to the number of neurons that make up the first FC layer. As a result, we can adjust the number of FC input neurons to find the optimal dimension for the given application.

To extract CNN-features as described above, the weights of the CNN that make up the C layers should be fixed in advance. To this end, transfer learning can be used [16]. In transfer learning, the cardinality of the target task's dataset is usually less than that of the source task's training dataset. Because of this, the proposed CNN-feature extraction method can avoid the general difficulty of CNN learning, which requires a large amount of data. Moreover, CNN training in the transfer learning mode is made up of fine-tuning. Therefore, the learning time can be significantly reduced with a small amount of fine-tuning.

The CNN model is learned by repeatedly performing forward propagation (FP) and backward propagation (BP), in an end-to-end manner. At this time, one of sum-, max-, or O2P-pooling is used to reduce the amount of processed data. When using O2P-pooling, the learning process is performed as follows: First, for the FP, it is assumed that a feature tensor, $X \in \mathbb{R}^{h \times w \times d}$, of the height, $h$, width, $w$, and channel, $d$, is generated from the last C layer, and that this tensor is reshaped into a matrix, $X \in \mathbb{R}^{n \times d}$, consisting of $n = hw$ features of the $d$-dimension. Then the second-order pooling, $\Sigma \in \mathbb{R}^{d \times d}$, for $X$ is computed as:

$$\Sigma = X \bar{I} X^{\mathrm{T}}, \tag{4}$$

where $\bar{I} = \frac{1}{n}\left(I - \frac{1}{n}\mathbf{1}\right)$; $I$ and $\mathbf{1}$ are the $n \times n$ identity matrix and the matrix of all the ones, respectively. To improve the representation power, $\Sigma$ is transformed into $Z$ by performing matrix power normalization using SVD (singular value decomposition) and is then sent to the FC layer.

Next, BP proceeds in the opposite direction. Given the gradient of a loss function, $l$ w.r.t. $Z$, propagated from the FC layer, i.e., $\partial l / \partial Z$, the gradient of $l$ w.r.t. $\Sigma$ is estimated sequentially. First, we have prepared all the relevant derivations (see Equation (2) in [31]),

and then we can calculate the $\partial l/\partial \Sigma$. Using this derivative, the gradient of $l$ w.r.t. $X$, which is used to update the weights of the network, is determined as:

$$\frac{\partial l}{\partial X} = \bar{I}X \left( \frac{\partial l}{\partial \Sigma} + \left( \frac{\partial l}{\partial \Sigma} \right)^{T} \right). \tag{5}$$

Here, it should be noted that the BP contains GPU-nonfriendly computations, such as SVD, which leads to costly training. To solve this problem, fast end-to-end methods were developed, such as MPN-COV [32] and iSQRT-COV [33], which are suitable for parallel implementation.

### 3.2. Seismic Data

A total of 500 synthetic seismic images were prepared, and each of them contained one fault line with different slopes and positions. The corresponding fault position in each image was indicated in white by generating binary masks, referring to the seismic amplitude information. Figure 1 presents an example of a seismic wave image (a), and a fault line (b) to be extracted from it. The dataset of the synthetic seismic wave images (501 × 501 pixels), reproduced through open-source code, IPF [37], is an artificial implementation of sequential rock deformation over time.

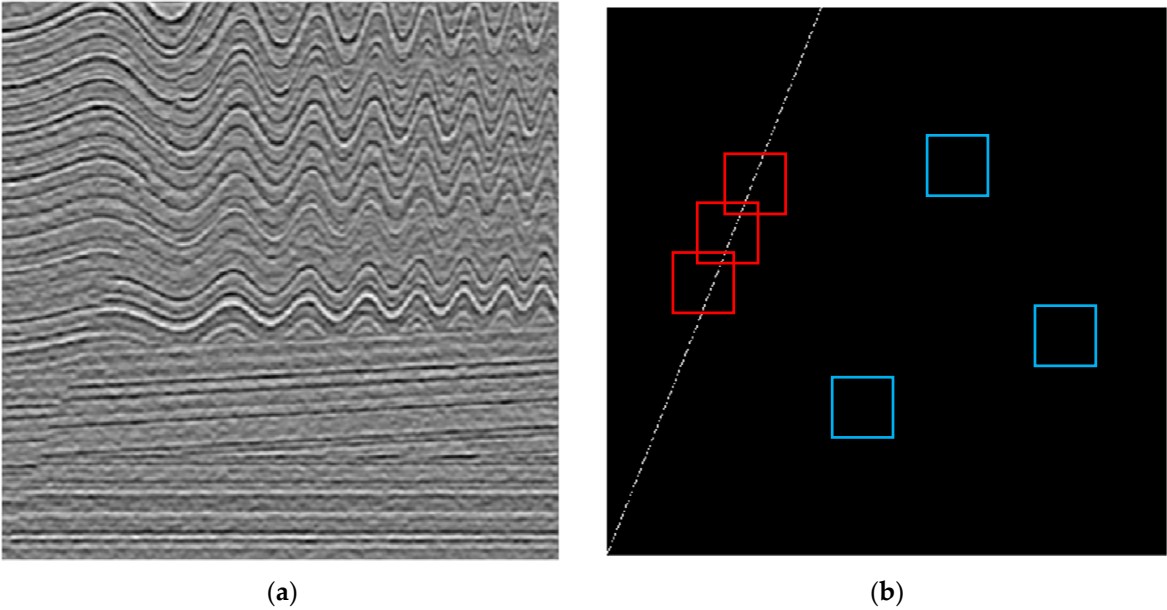

(**a**)　　　　　　　　　　　　　　　　　　　　　　　　　　(**b**)

**Figure 1.** Plots presenting synthetic seismic wave data: (**a**) a synthetic seismic image; (**b**) a fault line. Here, two blocks, marked in the red color and in blue, indicate the fault and nonfault patches, respectively.

From the dataset composed of image pairs, shown in Figure 1 (where the two images on the left and right sides include seismic waves and fault lines, respectively), the fault and nonfault patches were extracted. One patch is a (45 × 45)-dimensional matrix with one candidate pixel in the center, and 2024 pixels adjacent to it (this is the smallest patch size generating satisfactory results [4]). This patch can be classified as a fault patch or a nonfault patch, according to the following rule: If the candidate pixel is a pixel forming a fault line, then it becomes a fault patch; otherwise, it becomes a nonfault patch. For example, referring to the fault line matrix shown in Figure 1b, all possible fault patches were first extracted, and then the same number of nonfault patches were randomly extracted from the seismic wave image, shown in Figure 1a.

## 4. Experimental Study

In this section, we present the evaluation results for the classification performance and data complexity of the CNN-features. These were compared to those of the feature vectors, extracted in three ways: PCA; kPCA using "Gaussian" kernel [38]; and discriminative auto-encoder (AE) [39]. kPCA and AE were selected as the nonlinear and the network-based version of the PCA, respectively, and the implementations of the original authors were used without modification. Regarding CNNs, two versions were implemented [40]. The first is a typical CNN with sum-pooling, and the second is an improved CNN using O2P-pooling. Then, the SVMs were realized from the LIBSVM [41].

### 4.1. Synthetic and Real Experimental Data

As mentioned above, we first generated synthetic seismic waves and fault images. From the total of 500 seismic wave images, two patch sets of 'Test1' and 'Train1' were constructed as follows: Test1 is a set of 76,038 patches extracted from the first 100 images; and Train1 is a set of 284,850 patches extracted from 400 seismic images from the remaining 101 to 500 images. Train1 (and Test1) was used to pretrain the CNN models needed to extract the CNN-features.

For real-world seismic images, we considered the real seismic wave image data, cited from the Project Netherlands Offshore F3 Block—Complete [42]. We first asked some experienced experts to manually mark the fault lines according to the structural information (the work of assigning fault lines to real seismic wave images should be done by human labelers using appropriate tools, as was performed in ImageNet [28]). Next, for a simple comparative analysis, only ten seismic images along with fault lines were selected. Then, a total of 52,026 (26,013 for each class) fault and nonfault patches were extracted from the ten seismic images. When extracting the patches, the number of patches on the larger side was adjusted to the smaller side by randomly selecting them in order to prevent an imbalance between the classes. Hereafter, this real-world patch data is referred to as "RealPatch".

### 4.2. CNN Models Implemented

Figure 2 and Table 1 show the details of the CNN architecture designed for the experiment. This CNN consists of two convolutional (and subsampling: SUB) layers, and one FC layer, which is one of the smallest required scales for our goal. The parameters involved in the CNN are listed in Table 1.

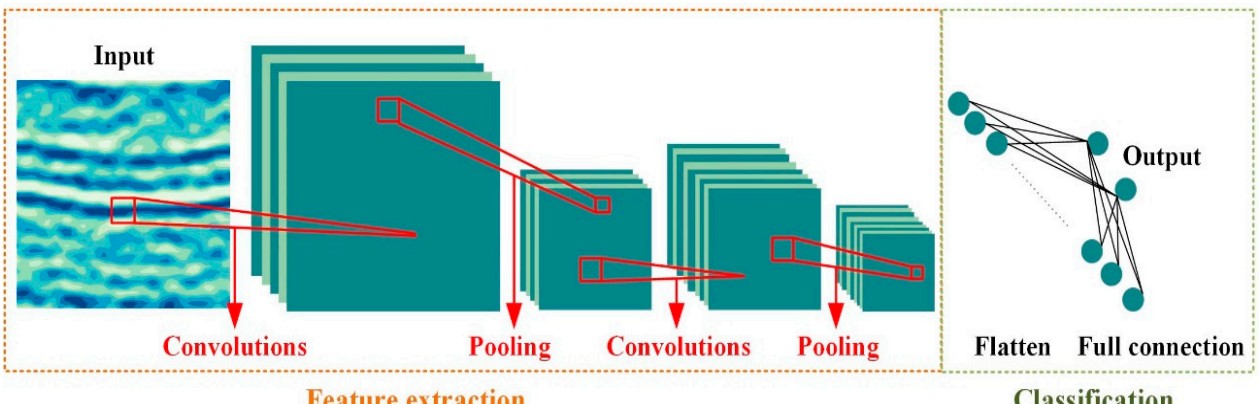

**Figure 2.** The architecture of the CNN model.

**Table 1.** The details of the CNN architecture.

| Layer Descriptions | Input Size | Output Size | Feature Maps | Kernel Size |
|---|---|---|---|---|
| 1st convolution and subsampling | $45 \times 45$ | $40 \times 40$ | 4 (8) | $6 \times 6$ |
| 2nd convolution and subsampling | $20 \times 20$ | $16 \times 16$ | 8 (16) | $5 \times 5$ |
| Fully connected | $512 \times 1$ | $2 \times 1$ | | |

In Table 1, the architecture is implemented with two different models, a conventional sum-pooling CNN and an improved O2P-pooling CNN, which are referred to as "CNN-sum" and "CNN-O2P", respectively. Details related to the models are as follows: In CNN-sum, the number of output maps of the two C layers was set to 4 and 8, respectively, but, in CNN-O2P, the number was adjusted to 8 and 16, as indicated by the parentheses in the fourth column of the table.

Second, in the two SUB layers, sampling was performed by sum-pooling, which reduces the dimension of each axis by half. At this time, the CNN-sum directly connected the output (which was reshaped to a vector) of the second SUB layer to the FC layer, but, in CNN-O2P, O2P-pooling was additionally implemented, and the obtained result was connected to the FC layer. In all these operations, pooling was performed with a stride of 2.

Third, the last component at the bottom of both models is the FC layer, and the number of input neurons of this layer is defined as the number of weights output from the previous layer. The CNN-features we are trying to extract consist of these input neurons that connect directly to the FC layer. Therefore, the dimension of the feature vector can be determined by adjusting the number of these neurons.

Fourth, each model deals with the two-class problem of classifying the seismic image patches as normal and abnormal. Therefore, it is necessary to adjust the number of neuron units in the output layer when expanding to multiclass problems. In addition, a soft-max function is required to obtain the normalized probabilities for each class.

Finally, the parameters for learning the CNN models were experimentally set to a learning rate of 1, a momentum of 0.5, a batch size of 10, and the number of epochs was set to 100. The number of epochs was optimized by referring to the learning curve to prevent overfitting.

*4.3. Classification Accuracy Rates*

Our CNN-features were extracted as follows: We first prepared a CNN model using source training data, Data1 (# of epochs: 100). Next, we fine-tuned this pretrained model using target training data, Data2 (# of epochs: 200). Here, as for Data1, Train1 was used as it was, and Data2, of 10,000 patches (5000 per class), was randomly selected from RealPatch.

In classification work performed by a feature extractor and a classifier, in general, there is no optimal feature extractor for all classifiers, and vice versa. Thus, to validate the performance of our CNN-based extractor, we conducted a simple classification experiment using only two types of classifiers: the k-nearest neighbor rules (kNNs: k = 1, 3), and SVMs (of the polynomial and RBF kernels). (Here, each of these two classifiers was chosen as the easiest to implement and the most widely used classifiers. Although kNNs have a long execution time, the classifiers were chosen because the discriminative characteristics of the input vectors can be directly reflected in the output determination. Therefore, thorough evaluation using other classifiers, such as AdaBoost and the decision tree, is a future challenge).

Table 2 presents a numerical comparison of the classification accuracy rates (%) of the features extracted by the five methods, i.e., PCA, kPCA, AE, CNN-sum, and CNN-O2P. Here, the values were averaged after repeating 10 times. Each time, 10,000 (and 10,000) patches were randomly selected as the training (and test) data from RealPatch, and the five different types of feature vectors were extracted. For a fair comparison, the dimensions of

all the feature vectors were adjusted to 256. The highest (second) accuracy of each row is emphasized in bold (underlined).

**Table 2.** Classification accuracy rates (mean) (%).

| Classifiers (Types) | Feature Extraction Methods | | | | |
|---|---|---|---|---|---|
| | **PCA** | **kPCA** | **AE** | **CNN-Sum** | **CNN-O2P** |
| kNN (k = 1) | 92.47 | 93.10 | 86.33 | <u>95.67</u> | **99.03** |
| kNN (k = 3) | 84.52 | 86.56 | 74.59 | <u>93.19</u> | **98.47** |
| SVM (polyn.) | 80.20 | <u>86.69</u> | 72.69 | 85.88 | **96.21** |
| SVM (RBF) | 91.16 | 93.32 | 83.58 | <u>95.91</u> | **97.99** |

From the comparison shown in Table 2, it can be observed that the use of the CNN-based extractor, in conjunction with existing classifiers, can significantly improve the classification accuracy compared to the other extractors included in the comparison. In particular, it is noteworthy that the CNN-based extractor uses not only training data, but also pretrained CNN models that could not be used for other extractions. Under this condition, a direct comparison of the five feature extractors may not be fair. However, from the perspective of machine learning, such as transfer learning, the results of this experiment suggest one possibility related to the "new" extractor. From this consideration, the performance improvements observed in the last column of the table may have resulted from the discriminating ability of the pretrained models.

Finally, in Table 2, CNN-sum and CNN-O2P were included as feature extractors, but they can also serve as classifiers with the FC layers. The classification performances of the CNN-sum and CNN-O2P trained for the experiments in the table were 96.13 (%) and 96.61 (%), respectively.

*4.4. Data Complexity Measures*

To highlight the reason why the use of the CNN-features improves the classification performance, the complexity measures were also measured and compared. Figure 3 shows a comparison of the complexity measurements obtained by applying the four extractors of PCA, kPCA, AE, and a CNN (CNN-O2P or CNN-sum) to RealPatch, prepared for this experiment.

In Figure 3, the CNN (CNN-O2P or CNN-sum) bar has the highest height for the $F1$ complexity, while the PCA (or kPCA) bar has the highest for the $Fv$, $D2$, and $L2$ complexities. Through this comparison, we can consider as follows: First, the result of comparing the data complexity shown in Figure 3 is consistent with the result of comparing the classification accuracy shown in Table 2. That is, compared to the accuracy of PCA and CNN-features in Table 2, CNN-based is superior to PCA. In Figure 3, when comparing the $F1$ values for these two extractors, the CNN bar height is higher than that of PCA.

Second, a comparison of the $Fv$, $D2$, and $L2$ complexities is the opposite of $F1$: the CNN-based bar height is lower than the PCA height. Specifically, the fact that the CNN-based accuracy rate in Table 2 is greater than that of the PCA is consistent with the fact that the CNN-based $F1$ bar in Figure 3 is higher than that of the PCA. However, the $Fv$ value is small when the accuracy rate is large. As reported in the relevant literature [13], the increasing value of $F1$ reduces the overlapping feature space and, thus, allows for a better separation of the feature space of the two classes. Unlike $F1$, the smaller the value measured in $Fv$, the simpler the classification problem.

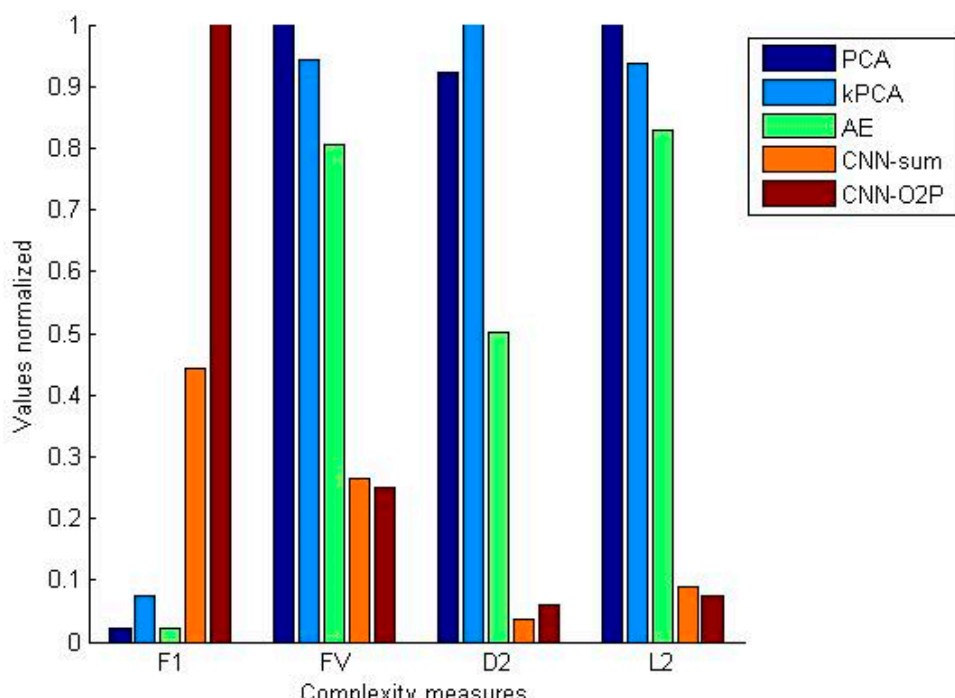

**Figure 3.** Plot comparing the complexity measures obtained in PCA, kPCA, AE, CNN-sum, and CNN-O2P from RealPatch. For ease of comparison, each of the complexity values measured was adjusted so that the maximum value was 1. In addition, CNN-sum and CNN-O2P were clearly compared only in F1. Thus, to clarify the focus of the paper, the two were analyzed only with a CNN.

Similar comparative analyses can be applied for the other complexity measures, $D2$ and $L2$. In general, for $D2$ and $L2$, the larger the bar height, the greater the overlap between classes, which makes classification difficult. The results of Figure 3 are very consistent with the above fact. From the observations made above, it can be argued that the CNN-based extractor is a good feature extractor when compared to the AE-based extractor, as well as when compared to the PCA (and kPCA). In particular, the extractor can be applied to situations where existing extractors do not work well because of the nature of the data. In various practices dealing with high-dimensional data, PCAs, which rely on covariance matrices, are known to be rarely used because they are not efficient.

*4.5. Time Complexity*

Finally, to make the comparison complete, the time complexity of the extraction methods was explored. Rather than embarking on another analysis of the computational complexities, however, the time consumption levels for the datasets were simply measured and compared. In the interest of brevity, the processing CPU times for each dataset is the time consumed by repeating the feature extraction several times and then averaging it. Table 3 presents a numerical comparison of the processing CPU times. Here, the times recorded are the required CPU times on a laptop computer with a CPU of 2.60 GHz and a RAM of 16.0 GB, and that is operating on a Window 64-bit platform.

**Table 3.** The processing CPU times (in seconds).

| Data | PCA | kPCA | AE | CNN |
|------|-----|------|-----|-----|
| Train1 | 8.3 | 7775.0 | 9560.8 | 35.5 |

In Table 3, it can be observed that the extractor of the CNN-features requires a much shorter processing CPU time than the traditional nonlinear algorithms for the datasets used. Particularly, this demonstrates that the extractor can dramatically reduce the processing

time compared to other kernel- and network-based methods (e.g., kPCA and AE). However, it should be noted that extracting the CNN-features requires a pretrained model. In Table 3, Train1 was used for the seismic data to obtain the pretrained model (# of epochs: 100). The training time for the model was excluded when counting the processing time in the table.

*4.6. Summary and Future Challenges*

As a result, from the above experiment, we can observe that the classification of seismic patch data can be improved by means of hybridizing CNN-features with existing classifiers. The summary of the experimental results and future tasks are as follows.

In this approach, CNN models were first trained to extract CNN-features. To this end, we first prepared a CNN model using synthetic data and then used it as a pretrained model when learning models for the feature extraction of real-life data. Therefore, it is necessary to analyze the extent to which such transfer learning has affected the CNN-feature extraction process.

More specifically, the task of analyzing the impact of the learning results of the source domain (i.e., CNN models learned from the synthetic data) on the classification performance of the target domain is a future challenge. We also experimented with a very limited number of real-world data here. Therefore, the task of investigating the optimal cardinality of the training set required for the target domain is also an open problem.

It is also interesting to see what has happened when using enhanced CNNs to extract CNN-features with the hybrid method. Here, however, only CNN-sum and CNN-O2P were implemented in one simple CNN and its weakly enhanced CNN, respectively, and were compared to each other. Therefore, experiments on strongly enhanced CNNs, including LesNet [43] and VGG-Net [44], remain open.

In order to find out why this hybrid approach leads to improved results, we measured the data complexity of the CNN-features and compared it with that of the existing extractors. However, complexity measurements were limited here, using only a limited number of measurements. In addition, the feature-extraction performance was compared using only a few traditional extractors. Therefore, introducing more diverse complexity metrics and including handcrafted extractors in comparison, such as the LBP, the HOG, etc., is a future challenge.

## 5. Conclusions

In this paper, the CNN-feature was first extracted using one CNN model, and was then used to learn an existing classifier, such as an SVM. Here, the model was first pre-trained using synthetic seismic data and was then fine-tuned using real-world data. By measuring the data complexity, the answer to the question of why this approach works effectively is presented.

However, the simulation was simply performed, by referring to the smallest structure that can express basic algorithms. Therefore, a comprehensive evaluation, using various architectures, such as ResNet and VGG-Net, and a comparison with the latest results in this domain remain a challenge in the future. In addition, detecting the occurrence of negative transfer that may occur in this approach is an important topic for future work. In addition to these limitations, the problem of theoretically investigating the CNN-based extractor remains unresolved.

**Author Contributions:** Software, methodology, conceptualization, formal analysis, funding acquisition, C.Z.; software, methodology, investigation, formal analysis, X.W.; software, methodology, formal analysis, Writing—original draft, S.-W.K. All authors have read and agreed to the published version of the manuscript.

**Funding:** This work was supported, in part, by the National Key Research and Development Program of China (No. 2018AAA0102201).

**Informed Consent Statement:** Written informed consent has been obtained from the patient(s) to publish this paper.

**Data Availability Statement:** Data associated with this research are available from the corresponding author.

**Acknowledgments:** The work of the third author was conducted while visiting Xi'an Jiaotong University, China. The authors appreciate the providers of the software and the related data used in this study, and, especially, Dave Hale for making the IPF codes available online. We also appreciate the hard work of the anonymous referees. Their detailed comments greatly improved this paper.

**Conflicts of Interest:** The authors declare that they have no known competing financial interests or personal relationships that could have appeared to influence the work reported in this paper.

## Glossary

| Abbreviation | Meaning |
|---|---|
| AE | auto-encoder |
| B-CNN | bilinear convolutional neural network |
| BP | backward propagation |
| BiP | bilinear pooling |
| C | convolutional |
| CNN | convolutional neural network |
| FC | fully connected layer |
| FP | forward propagation |
| GPU | graphics computing units |
| HOG | histogram of oriented gradients |
| HYB | hybrid method of combining linear (nonlinear) classifier with CNN |
| iSQRT-COV | iterative matrix square root normalization of covariance pooling |
| LBP | local binary pattern |
| MPN-COV | matrix power normalized covariance pooling |
| O2P | second-order pooling |
| PCA | principal component analysis |
| PR | pattern recognition |
| SVM | support vector machine |

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
