# Peer review of "Empirical Evaluation on Utilizing CNN-Features for Seismic Patch Classification"

_applsci, doi:10.3390/app12010197_

Round 1
Reviewer 1 Report
The paper proposes a novel CNN-features for seismic patch classification. Then the CNN-features are used by different classifiers to calculate the accuracy rates. The models with CNN-featrues outperform other existing extracted features due to the power of pretrained CNN extractor. Overall I think this paper worth publishing. However, some points need to be addressed:
- In the Related Work section, the authors include the model improvement works of CNN, but no related work about previous seismic patch classification is cited. Please add more related work on this to help readers understand the context.
- "The use of SVMs instead of FC layers is known to improve classification accuracy [7], [12], but no analysis has been made on why. Rather than embarking on a general analysis,in this paper we will consider a comparative study that will be taken as the basis for the above improvements. " - The authors claim to perform a comparative study for those clasiffiers, but only kNN and SVM models are compared. Why the FC is excluded here? Please clarify on this.
- Figure 2's font size is a little bit small. Might consider increasing the font size and resolution.
Author Response
Reviewer #1:
1) In the Related Work section, the authors include the model improvement works of CNN, but no related work about previous seismic patch classification is cited. Please add more related work on this to help readers understand the context.
Answer: The authors agree with the reviewer's comments. Subsection 2.1 is newly added to review the related literature on seismic fault detection. Considering that the topic discussed in the articles recommended by the Reviewer are far away from our research, we added some closely ones in the paper (refer to the relevant contents of the text marked "blue").
2) "The use of SVMs instead of FC layers is known to improve classification accuracy [7], [12], but no analysis has been made on why. Rather than embarking on a general analysis,in this paper we will consider a comparative study that will be taken as the basis for the above improvements. " - The authors claim to perform a comparative study for those classifiers, but only kNN and SVM models are compared. Why the FC is excluded here? Please clarify on this.
Answer: Thanks for the Reviewer's valuable comment. The comparison in Table 2 is related to the performance of feature-extractors such as PCA, kPCA, AE, and CNN-based methods rather than classifiers. That is, CNN-sum and CNN-O2P are CNN models from which feature-vectors are extracted, and thus, it is not fair to compare the two CNN models in the same table. Therefore, to clarify why the CNN-based classifiers were excluded from the comparison, we revised the description of Table 2 and added a new description.
3) Figure 2's font size is a little bit small. Might consider increasing the font size and resolution.
Answer: The subplots shown in Figs. 1-2 have been made more clearly. In addition, we further improved the English writing of the whole paper, and also corrected some typographical and grammatical errors.
Reviewer 2 Report
The paper presents a study about an improved CNN for identifying seismic patch into images. The paper is also well-written and presented, but in this reviewer's opinion ahoturs should pay attention to some details. Please, see the comments in the attached PDF.

Author Response
Reviewer #2:
The authors would like to thank the referee for reviewing our paper so carefully and providing us much valued feedback. The answer to each comment begins with Answer:
(1) In this section (and in the introduction), authors provide the aspects treated in this paper for improving the use of CNN, trying to overcome the actual limits. Nevertheless, in this reviewer opinion, this section completely misses of an overview about the use of ML techniques (and CNN and transfer learning) in the field of structural and seismic engineering, such as the aspect herein faced. Some additional examples and scopes should be mentioned, such as the ones reported in https://doi.org/10.1016/j.autcon.2021.103936; 10.1177/8755293019878137
Answer: Thanks for the Reviewer's valuable comment. The authors agree with the reviewer's comments. Subsection 2.1 is newly added to review the related literature on seismic fault detection. Considering that the topic discussed in the articles recommended by the Reviewer are far away from our research, we added some closely ones in the paper (refer to the relevant contents of the text marked "blue").
(2) Please, insert a proper equation.
Answer: We rewrote the several important formulas as shown in Eqs. (1-5).
(3) what does it mean? Please, report the extended name
Answer: We revised the sentence to clearly indicate what HYB means.
(4) Are they enough as starting DB?
Answer: The data set of this paper contains more than 360,000 patches extracted from 500 seismic images. Using these data, we successfully trained the CNN architecture described in Figure 2 and Table 1. This point was clarified in the revised manuscript.
(5) Honestly, Figure b should be more clear
Answer: The subplots shown in Figs. 1-2 have been made more clearly.
(6) For all websites inserted in the text, use a specific reference and put the websites in the proper reference section
Answer: We added all websites as references according to the request for revision.
(7) Did you train an existing architecture or did you made a new net architecture? Could you add a figure about it?
Answer: In Fig. 2, we added a new plot to illustrate the architecture of the CNN.
(8) Is there a specific code or a specific path to use the proposed approach? It should be mentioned for reproductibility scopes. In addition, could you insert a part in which you demonstrate the efficiency of your proposal, such as a validation?
Answer: To provide details regarding where data supporting reported results can be found, we newly added the data availability statement in this revised version.
Reviewer 3 Report
The manuscript entitled “Empirical Evaluation on Utilizing CNN-features for Seismic Patch Classification” investigated a CNN-based model for seismic patch image classification.
After careful review, the manuscript has a reasonable effort and technical information. Firstly, I am not sure by having such a high accuracy just by CNN and fundamental optimization about the accuracy of the data and applicability of the method. It has to be clarified in the next response letter. However, there is not much novelty on the work, and there are some points that must be considered in the revision to be worth being accepted. Therefore, I strongly recommend the authors to follow the comments below:
1- As you have many abbreviations, therefore please provide an abbreviation table that on MDPI format must be at the end of the paper, however you made it but is not complete.
2- Abstract: The sentence is a bit confusing and is very long. You can concise it and point the important parts.
3- The work presents a poor literature review on the methods based on soft computing (ML, Fuzzy and AI) for seismicity assessment and earthquake damage estimation. Because your work is not much new and novel, I strongly recommend you make your introduction and study on previous methods interesting for readers by adding the following new works which I found it new and related to your work which are based on different methods for rapid damage assessment of buildings. It will increase the dept of your review and stronger and wider area.
-Multi-Hazard and Spatial Transferability of a CNN for Automated Building Damage Assessment
-A review on application of soft computing techniques for the rapid visual safety evaluation and damage classification of existing buildings
-A Synthesized Study Based on Machine Learning Approaches for Rapid Classifying Earthquake Damage Grades to RC Buildings
-Bai, Y.; Hu, J.; Su, J.; Liu, X.; Liu, H.; He, X.; Meng, S.; Mas, E.; Koshimura, S. Pyramid Pooling Module-Based Semi-Siamese Network: A Benchmark Model for Assessing Building Damage from xBD Satellite Imagery Datasets. Remote Sens. 2020, 12, 4055.
-Nex, F., Duarte, D., Tonolo, F. G., & Kerle, N. (2019). Structural building damage detection with deep learning: Assessment of a state-of-the-art cnn in operational conditions. Remote sensing, 11(23), 2765.
-CABiNet: Efficient Context Aggregation Network for Low-Latency Semantic Segmentation
-Shin, H. C., Roth, H. R., Gao, M., Lu, L., Xu, Z., Nogues, I., ... & Summers, R. M. (2016). Deep convolutional neural networks for computer-aided detection: CNN architectures, dataset characteristics and transfer learning. IEEE transactions on medical imaging, 35(5), 1285-1298.
-Valentijn, T. (2020). The Practical Applicability of a CNN for Automated Building Damage Assessment.
4- Figure 1 and 2: Please improve the quality and font size to be more visible
5- Please provide a figure that shows the architecture of your model and network. It needs more description and make it more understandable.
6- Please make your tables and figures follow a same path and font size and colors
7- There are many typos that needs to be corrected for instance spaces after (.) amd (,) which in some cases are doubled or missed.
8- Generally, it seems that you have a very high accuracy and it is a bit strange because you did not present the ROC of each class and did not control the overfitting. Especially in the case of using different data for training and test! Please clarify more in this manner.
9- You did not highlight the problem statement, objectives and novelty of your proposed method.
10- There is a need in proofreading the work.
11- Please provide more information about the data collection or case study if they are from some specific region. I recommend to sumarize the lable of figure and write more in the body of the text.
12- Please provide more information about the analysis rather than just F1 score. Show your confusion matrix and the parameters you can achieve from it.
At the end as I have mentioned, there are not much significant novelty on this work but the efforts were good and it would be good if you revise it according to the points provided and other reviewers.
Author Response
Reviewer #3
At present, we have revised our paper carefully according to the constructive comments suggested by the Reviewers. The main changes we have made are summarized as follows.
1) Some sentences in Abstract have been reformulated to avoid any confusions.
2) Subsection 2.1 is newly added to review the related literature on seismic fault detection. Considering that the topic discussed in the articles recommended by the Reviewers are far away from our research, we added some closely ones in the paper.
3) As for the several important formulas, we rewrote them as shown in Eqs. (1-5).
4) The photos shown in Figs. 1 and 3 have been replaced with new ones to make the contents clearer.
5) In Fig. 2, we added a new plot to illustrate our model and the architecture of the CNN.
6) The comparison in Table 2 is related to the performance of feature-extractors rather than classifiers. Therefore, to clarify why the CNN-based classifiers were excluded from the comparison, we revised the description of Table 2 and added a new description.
7) All figures and tables are following the template provided by the Journal.
8) At the end of the paper, a table is added to show all abbreviations and their full names to facilitate the reading of the paper.
9) Finally, we further improved the English writing of the whole paper, and also corrected some typographical and grammatical errors.
In a word, thank you very much for your valuable suggestions and providing us an opportunity to improve our paper.
Round 2
Reviewer 3 Report
Dear Authors
Many thanks for your responses and significant changes.